

# Evaluating the efficacy of AI-driven intrusion detection systems in IoT: a review of performance metrics and cybersecurity threats

Jianwei Tian and Hongyu Zhu

Hunan Key Laboratory for Internet of Things in Electricity, State Grid Hunan Electric Power Company Limited Information and Communication Company, Changsha, China

## ABSTRACT

**Background:** The growing scale and complexity of Internet of Things (IoT) environments have intensified the need for intelligent and adaptive cybersecurity mechanisms. Artificial intelligence (AI)-based intrusion detection systems (IDS) have emerged as a promising solution for identifying and mitigating threats in real time.

**Methodology:** This review systematically evaluates the effectiveness of AI-based IDS in IoT networks, following the Preferred Reporting Items for Systematic Reviews and Meta-Analyses (PRISMA) 2020 guidelines. A comprehensive search of the Scopus and Web of Science databases was conducted, yielding 203 studies, of which 51 met the inclusion criteria. Eligible studies, published between 2016 and 2025, were analyzed for geographic distribution, AI techniques used, methodological quality, and reported outcomes. Meta-regression and contour-enhanced funnel plots were employed to assess effect size trends and publication bias.

**Results:** Most studies originated from India, Saudi Arabia, and China, with research output peaking in 2024. Meta-regression analysis revealed a positive correlation between publication year and reported effect size, indicating progressive advancements in AI methodologies. Machine learning (ML) and deep learning (DL) were the most widely used techniques, with a growing trend toward hybrid and ensemble models that enhance threat detection accuracy. Recent studies also showed increased interest in explainable artificial intelligence (XAI), reflecting the demand for transparency and interpretability in model outputs. Funnel plot asymmetry suggested a bias toward publishing positive findings.

**Conclusions:** AI-based IDSs have demonstrated substantial potential in strengthening IoT security, particularly through ML, DL, and hybrid approaches. However, inconsistencies in evaluation metrics, reporting standards, and methodological design remain significant challenges. The findings highlight the need for standardized benchmarks and robust frameworks to guide future research and ensure reliable deployment of AI-driven IDS in diverse IoT contexts.

Corresponding author
Hongyu Zhu, jack3330515@163.com

## INTRODUCTION

The rapid expansion of the Internet of Things (IoT) has transformed the digital landscape by enabling seamless connectivity across a vast range of devices, applications, and services. From industrial automation and smart cities to personal health monitoring and home automation, IoT is increasingly embedded into the fabric of daily life (*Hasan, Moon & Raza, 2023*). This evolution has greatly enhanced the intelligence of manufacturing processes through the integration of advanced technologies such as artificial intelligence (AI), cloud and edge computing, big data analytics, robotics, and cybersecurity (*Arslan et al., 2024*). The very nature of IoT marked by its distributed architecture, limited-resource devices, and heterogeneity introduces serious vulnerabilities. These inherent weaknesses have made IoT systems highly susceptible to a wide array of cybersecurity threats, ranging from denial-of-service (DoS) attacks and spoofing to botnets and data breaches (*Alrayes et al., 2024*). IoT connects billions of devices, generating an immense volume of data, from megabytes to geopbytes. Cisco predicted over 50 billion connected devices by 2020, highlighting massive traffic and data flow. About 40% of this data is processed near the network edge due to cloud limitations in meeting IoT demands (*Almiani et al., 2020*). Traditional intrusion detection systems (IDSs), typically rule-based or signature-based, struggle to cope with the complex, dynamic, and large-scale nature of IoT environments (*Kulrujiphat & Kulrujiphat, 2024*). The rising complexity of attacks like brute-force, malware, and phishing threatens data security and business continuity. Traditional IDS face challenges such as inefficient feature selection, high false positives, and limited scalability. These limitations are especially critical in resource-constrained IoT environments (*Lella et al., 2025*). AI-IDSs can autonomously learn patterns, adapt to emerging threats, and operate more effectively in complex and data-rich environments like IoT networks (*Benaddi et al., 2022*).

Despite growing interest in AI-based IDSs, the literature remains fragmented and inconsistent. Variations in datasets, threat models, evaluation metrics, and methodologies hinder the ability to draw generalizable conclusions about AI-IDS effectiveness in IoT environments. Many studies rely on controlled settings or benchmark datasets that may not reflect real-world complexity, and inconsistencies in reporting metrics like accuracy, precision, recall, F1-score, and false positive rate without a standardized framework further obscure meaningful comparisons (*Ahanger, 2018*; *Gueye et al., 2023*; *Konda, Ayyannan & Chandramouli, 2023*; *Saleh et al., 2025*). While systematic reviews have offered valuable insights, they often emphasize narrow threat categories, isolated performance indicators, or single algorithmic approaches. *Abdullahi et al. (2022)* focuses on AI-based methods for IoT cybersecurity, examining machine learning (ML)/deep learning (DL) techniques, IDS architectures, and their effectiveness across various attack scenarios. It finds that support vector machine (SVM), random forest (RF), and DL models achieve high detection accuracy. The review of *Ali, Khan & Khalid (2023)* present how ANNs enhance IoT security, addressing threats like DoS, distributed denial-of-service (DDoS), and intrusions through IDS frameworks and other mechanisms. It finds that ANN-based models significantly strengthen IoT security. To address these limitations, a rigorous

meta-analytical synthesis is needed. Meta-analysis provides a quantitative aggregation of findings across studies, enabling more robust and generalizable insights into the performance and applicability of AI-IDSs. Specifically, this review aims to: (1) identify prevailing AI methodologies utilized in IDS, (2) classify key performance metrics and threat categories, and (3) identify critical gaps in current research especially the lack of standardized evaluation frameworks and holistic threat coverage. By integrating diverse perspectives, this study contributes to the evolving discourse on IoT security, offering actionable insights for researchers, developers, and cybersecurity practitioners. It lays the groundwork for future research to refine methodologies, prioritize effective detection strategies, and address emerging threats in AI-driven cybersecurity.

This article is structured as follows. The next section critically evaluates existing literature on AI-driven IDSs in IoT environments and outlines the rationale for this meta-analytical review. The methodology describes the systematic review process, including database selection, search strategy, inclusion/exclusion criteria, data extraction, and quality assessment. The results and discussion present key findings on study characteristics, dominant AI techniques, performance metrics, and addressed cybersecurity threats. The research gaps section identifies methodological limitations, underexplored threats, and the need for standardized frameworks. The conclusion summarizes theoretical insights, practical implications, and future research directions.

## RELATED WORK

Research on AI-driven IDS within IoT environments has expanded rapidly with numerous studies employing machine learning (ML), deep learning (DL), and hybrid techniques to enhance security. The existing body of review literature generally falls into three key categories, those focusing on IDS architectures and AI model types, those centered around specific datasets or attack vectors, and broader overviews addressing general IoT security issues as shown in Table 1. Several systematic reviews have explored AI model selection and IDS architecture in detail. For example, *Mallidi & Ramisetty (2025)* analyzed 54 studies highlighting centralized, distributed, and federated learning models across cloud, fog, and edge computing layers. *Abdullahi et al. (2022)* reviewed 80 articles utilizing various ML algorithms such as SVM, RF, eXtreme Gradient Boost (XGBoost), neural networks, and RNNs to assess their efficacy in identifying IoT-specific threats. *Ali, Khan & Khalid (2023)* conducted an extensive review of 143 studies focused on artificial neural network (ANN)-based approaches, showcasing their integration into IDS systems and their contributions to cybersecurity. Despite their thoroughness, these reviews largely overlook considerations such as scalability, energy efficiency, and the development of lightweight hybrid models suitable for resource-limited, real-time applications.

A second research stream concentrates on dataset-specific and attack-targeted IDS strategies. The study of *Sana et al. (2022)* reviewed 41 articles that emphasized data preprocessing and feature selection for anomaly detection particularly using deep and reinforcement learning models. *Sejaphala, Malele & Lugayizi (2024)* offered a targeted analysis of 17 studies dealing with routing attacks in RPL-based IoT networks focusing on ML-based detection methods and their accuracy. *Mishra & Pandya (2021)* reviewed 185

**Table 1 Comparison of related review studies.**

| Study | Review type | Number of reviewed studies | Aim | Focus | Gaps |
|---|---|---|---|---|---|
| *Mallidi & Ramisetty (2025)* | Systematic literature review | 54 | To explore advancements in training and deployment strategies of AI-based IDS in IoT, evaluate their effectiveness, and propose future research directions. | Emphasis on IDS architectures, AI techniques (ML/DL), training paradigms (centralized, distributed, federated), deployment layers (cloud, fog, edge), datasets, and performance metrics. | Limited discussion on real-time adaptability, energy efficiency of training models, and integration challenges in resource-constrained IoT environments. |
| *Salem et al. (2024)* | Comprehensive review | 60 | To assess the effectiveness and limitations of AI techniques in detecting and preventing a broad spectrum of cyber threats. | Comparison of ML, DL, and metaheuristic algorithms in handling cyberattacks (*e.g.*, malware, intrusions, spam), along with a proposed evaluation framework. | Lack of empirical validation across diverse attack scenarios; minimal exploration of hybrid or ensemble learning techniques for adaptive threat detection. |
| *Abdullahi et al. (2022)* | Systematic literature review | 80 | To categorize, map, and assess existing AI methods for detecting cybersecurity threats in IoT settings. | Analysis of AI techniques (*e.g.*, SVM, RF, XGBoost, NN, RNN), smart IDS frameworks, attack detection effectiveness, and future research directions. | Absence of comparative studies on computational efficiency, model scalability, and explainability in AI-based IDS; lack of benchmarks for cross-evaluation. |
| *Sana et al. (2022)* | Systematic review | 41 | To improve security mechanisms in IoT by analyzing data transformation techniques for anomaly detection using deep learning. | Exploration of datasets, preprocessing techniques, performance metrics, features, and models for anomaly detection in IoT, with emphasis on deep and reinforcement learning. | Inadequate focus on adversarial robustness, generalizability across datasets, and the role of unsupervised learning in dynamic IoT environments. |
| *Mishra & Pandya (2021)* | Systematic review | 185 | To assess existing IoT security risks, especially DDoS attacks, and explore current IDS models, challenges, and future solutions. | Analysis of DDoS attack types and mitigation techniques, classification of IDS, ML/DL-based anomaly detection, and future security challenges in IoT. | Underexplored integration of multi-layered defense strategies; limited focus on real-time mitigation and edge intelligence; challenges in standardizing IDS evaluations across diverse DDoS scenarios. |
| *Sejaphala, Malele & Lugayizi (2024)* | Systematic review | 17 | To compare traditional and advanced machine learning algorithms for detecting routing attacks in RPL-based IoT networks. | Performance analysis of ML techniques for IoT routing attack detection, highlighting accuracy, false positive rate, and algorithmic effectiveness (*e.g.*, Random Forest). | Insufficient exploration of lightweight ML models suitable for RPL-constrained devices; absence of attack adaptation models for evolving routing threats. |
| *Ali, Khan & Khalid (2023)* | Systematic review | 143 | To investigate how ANNs contribute to enhancing security in IoT systems. | ANN-based solutions for IoT security, including models, frameworks, IDS mechanisms, and performance across various security features. | Limited analysis of interpretability, training cost, and integration of ANNs with real-time IoT communication protocols; lack of unified frameworks combining ANN with other intelligent agents for enhanced resilience. |

works related to DDoS mitigation and anomaly-based IDS design for IoT. Although insightful in specialized contexts these reviews tend to overlook issues such as adversarial robustness, model generalization across datasets, and the development of adaptive IDS frameworks suited for evolving IoT ecosystems. The third category includes broad surveys, such as *Salem et al. (2024)*, which synthesized findings from 60 studies comparing ML, DL, and metaheuristic methods across various cyber threats. Their work proposed a general evaluation framework for AI-based IoT security, but lacked detailed empirical validation across diverse attack scenarios and paid limited attention to hybrid or ensemble approaches capable of supporting adaptive real-world threat detection.

Across these three thematic areas, several critical research gaps emerge. Firstly, the need for real-time adaptability is insufficiently addressed particularly in relation to latency and edge computing for prompt threat mitigation. Secondly, energy efficiency and resource sensitivity are underexplored, especially regarding lightweight models deployable on constrained IoT devices. Thirdly, there is a lack of standardized performance evaluation and model interpretability. Few studies offer quantitative syntheses to support operational deployment. This meta-analysis seeks to fill these gaps by quantitatively evaluating the performance of AI-based IDS solutions in IoT environments. It incorporates heterogeneity assessment, temporal performance trends, and bias diagnostics.

# METHODS

The Methods section outlines the procedures employed to retrieve relevant publications, along with the inclusion criteria and limitations applied to select eligible studies (*Amundsen et al., 2018*; *Ibrahim & Mahmoud, 2025*). It then details the standards used to evaluate and interpret the studies and their associated variables. This meta-analysis adheres to the Preferred Reporting Items for Systematic Reviews and Meta-Analyses (PRISMA) framework (*Page et al., 2021*), ensuring transparency and rigor. PRISMA guidelines offer a structured approach for conducting and reporting systematic reviews and meta-analyses. They promote methodological rigor, transparency, and reproducibility by providing clear procedures for identifying, screening, assessing eligibility, and selecting relevant studies. The primary goal of this methodology is to minimize potential biases, making it a vital element in ensuring the credibility and reliability (*Jasim et al., 2025*). A comprehensive checklist PRISMA is included to support the systematic review process in Fig. 1.

## Search strategy

The search strategy involved systematic queries across two major academic databases Scopus, and Web of Science. Both databases were selected due to their extensive repositories of scholarly literature, which support a thorough exploration of the research domain. The stringent indexing standards applied by these databases ensure the academic credibility and reliability of the included studies (*Martín-Martín et al., 2018*; *Pranckutė, 2021*). Keywords such as ("artificial intelligence" OR AI) AND ("intrusion detection system" OR IDS) AND ("internet of things" OR IoT) AND (efficacy OR performance OR accuracy OR precision OR recall OR "detection rate") AND ("cybersecurity threat" OR

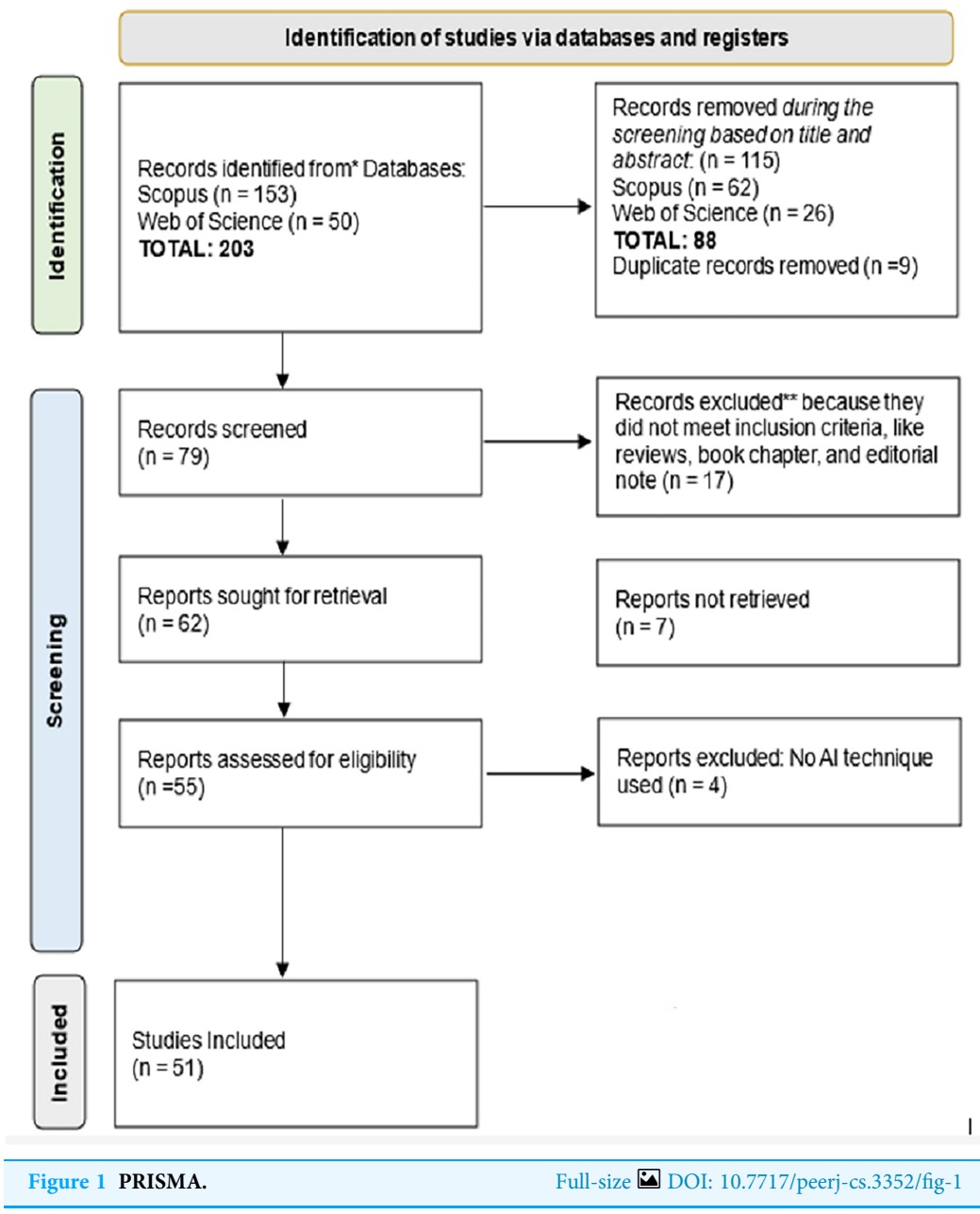

**Figure 1 PRISMA.**

"security attack" OR "cyber-attack" OR "cyber threats") were used in various combinations. The search was conducted in April 2024 to capture the most current and pertinent literature available at that time.

## Study inclusion and exclusion

Strict inclusion and exclusion criteria were employed to guarantee that only highly relevant studies were incorporated into this meta-analysis. The selection parameters used in this meta-analysis, as detailed in Table 2, were designed to enhance the accuracy, objectivity, and practical relevance of the findings by focusing specifically on research related to AI-based intrusion detection systems in IoT.

**Table 2 PICO.**

| Criteria | Inclusion | Exclusion |
|---|---|---|
| Focus of study | Studies on AI-based Intrusion Detection Systems (IDS) applied in IoT environments | Studies focusing only on traditional IDS or unrelated IoT security topics |
| Data reporting | Provides quantitative performance metrics (*e.g.*, accuracy, precision, recall, F1-score) | Lacks sufficient statistical or performance data for meta-analysis |
| Publication type | Peer-reviewed journal articles and conference articles | Grey literature (thesis, technical reports, white articles, preprints) |
| Publication period | Studies published between 2016 and 2025 | Studies published before 2015 or outside the target period |
| Language | Written in English | Non-English publications |
| Duplicates/ Redundancy | Most comprehensive version retained if duplicates or extended articles exist | Duplicate studies or less complete versions of the same research |

## Study selection

The study selection process followed the PRISMA 2020 guidelines to maintain rigor and transparency in identifying pertinent research on the efficacy of AI-driven IDS in IoT environments. During the identification phase, a comprehensive search of two major databases yielded a total of 203 records, with 153 from Scopus and 50 from Web of Science. In the screening phase, 88 records were excluded due to irrelevance, and nine duplicate records were identified and removed, leaving 79 records for full-text screening. During the eligibility assessment phase, 17 records were excluded for not meeting the basic inclusion criteria, and seven reports could not be retrieved. Of the 55 reports assessed for eligibility, four studies were excluded for not applying AI techniques. Ultimately, 51 studies were included in the final meta-analysis.

## Eligibility criteria (PICO)

The inclusion criteria were defined using the Population, Intervention, Comparison, Outcomes (PICO) framework, studies were included if they: (1) were published between 2016 to 2025, (2) were written in English, and (3) focused primarily on AI-based IDS techniques in IoT environments. Studies were excluded if they: (1) focused on unrelated technologies (*e.g.*, non-AI approaches or non-IoT contexts), or (2) lacked empirical or experimental data, theoretical articles, non-English articles, dissertations, and studies without performance metrics. This ensured relevance, methodological rigor, and comparability for the meta-analysis. Table 3 outlines the specific PICO elements applied in determining study eligibility.

## Data extraction

We used a structured Microsoft Excel spreadsheet to systematically extract key information from each study included in the review. The extracted data included the authors, year of publication, study design, IoT application domain, types of AI models used, feature engineering techniques, datasets utilized, and performance metrics such as accuracy, precision, recall, and F1-score. Additional details captured comprised threat scenarios addressed, adversarial strategies applied, comparisons with conventional methods, and the country of the study. Data extraction was conducted independently by

**Table 3 Comparative analysis of AI performance in IoT IDS.**

| Criteria | Descriptions |
|---|---|
| Population (P) | IoT systems and environments. |
| Intervention (I) | AI-based intrusion detection approaches. |
| Comparison (C) | Conventional IDS or other AI techniques. |
| Outcome (O) | Performance metrics such as accuracy, recall, F1-score, and false positives. |

two reviewers, with any disagreements resolved through discussion to ensure accuracy and reduce bias.

To handle variations in reported metrics, we applied a data harmonization strategy. Accuracy was chosen as the primary metric for quantitative comparison due to its consistent reporting across studies. Supplementary metrics such as F1-score, precision, and recall were analyzed when available. Studies that lacked at least one standard performance metric were included in the qualitative synthesis but excluded from the quantitative analysis. To account for variations in datasets and AI methodologies, we categorized studies by AI type and IoT application domain. This classification allowed for more meaningful subgroup comparisons rather than attempting a single aggregated analysis.

## Quality assessment

To maintain methodological rigor, we assessed the quality of each included study using a modified version of the Critical Appraisal Skills Programme (CASP) checklist (*Long, French & Brooks, 2020*). Evaluation criteria covered the clarity of research objectives, validity of datasets, completeness of performance metrics, soundness of experimental design, and relevance to real-world IoT environments. Studies were rated on a standardized scale, and only those meeting a predefined quality benchmark were included in the meta-analysis. This thorough vetting process ensured that only high-quality, credible studies informed our findings.

## Model taxonomy and classification

To maintain uniform terminology throughout the meta-analysis, a standardized taxonomy is established for classifying Intrusion Detection System (IDS) models. This classification differentiates between deep learning, ensemble, and hybrid approaches based on their structural design and methodological integration. The framework is applied consistently across the manuscript to ensure clarity and prevent misinterpretation Table 4.

# RESULTS

This section provides the demographic characteristics of the included studies and also result and discussion of the main findings.

## Study characteristics

The section discusses the demographic characteristics of the included studies based on annual publications and countries of the published included studies. Table 5 provides a

**Table 4 Model taxonomy and classification.**

| Category | Definition | Example models/Techniques |
|---|---|---|
| Deep learning | Models that learn hierarchical or temporal features from data automatically. | CNN, RNN, LSTM, GRU, autoencoder, transformer |
| Ensemble | Methods that combine multiple base learners to improve prediction robustness. | Random forest, XGBoost, bagging, boosting, stacking |
| Hybrid | Approaches integrating heterogeneous techniques (*e.g.*, supervised + anomaly detection or rule-based). | DL + signature-based, ML + statistical detector |

comprehensive summary of the 51 studies analyzed in this study, outlining the AI techniques employed, datasets utilized, and the specific cybersecurity threats addressed. This overview allows readers to discern recurring methodological trends and differences among the studies. Commonly used public datasets included CICIDS 2017 (*Indra et al., 2024*), UNSW-NB15 (*Keshk et al., 2023*; *Mousavi, Sadeghi & Sirjani, 2023*), and NSL-KDD (*Sadhwani et al., 2025*; *Tawfik, 2024*), while some research drew on proprietary or simulated IoT data. The primary focus areas were intrusion detection, DDoS attacks, and malware classification, highlighting prevailing concerns in AI-driven smart grid and IoT security. This table lays the groundwork for the detailed qualitative and quantitative analyses presented in the subsequent sections.

## Annual publications

The included studies are distributed across several years, with the highest number of studies occurring in 2024, where 20 studies were published. Following closely are 2023 and 2025, with 12 and 11 studies, respectively, reflecting ongoing research activity in recent years. The year 2022 also saw significant contributions, with six studies published. Fewer studies were recorded in 2018 and 2020, each with 1 study. This shows a clear increase in research output starting from 2022, peaking in 2024, and continuing strongly into 2025 as shown in Fig. 2.

## Countries of publications

The included studies are primarily from India, which stands out with 11 studies, making it the leading contributor. Saudi Arabia follows with seven studies, showing its strong involvement in research on IoT and cybersecurity. China is also well-represented with four studies, reflecting its active role in AI and intrusion detection research. Other countries like Jordan, Algeria, and Australia each contribute three studies, highlighting their regional importance in this area. Egypt, France, the USA, and Morocco each appear twice, suggesting a significant interest in the field from these nations. Countries such as Iran, the United States, the United Kingdom, Nigeria, Bangladesh, Bulgaria, Malaysia, Yemen, the UAE, Pakistan, Thailand, and Italy each have one study, showcasing the global scope of research in AI-driven intrusion detection systems as shown in Fig. 3. Among the 51 studies reviewed, a clear geographic concentration was observed with the majority originating from India, Saudi Arabia, and China. This prominence is largely due to these nations' significant research productivity in IoT-enabled smart grids and AI applications driven by

**Table 5 Summary of included studies.**

| Authors name | AI models used | Datasets used | Threat scenarios (Types of attacks) |
|---|---|---|---|
| Gueye et al. (2023) | MLP, CNN, RNN with embedding layer | ToN_IoT (Modbus subset), IoTID20 | Backdoor, injection, password, scanning, XSS |
| Saheed, Omole & Sabit (2025) | LSTM, attention mechanism | SWaT, WADI | Data pilfering, MitM, port-sweep, botnet attacks |
| Kulrujiphat & Kulrujiphat (2024) | Various (including ML and DL models) | Edge-IIoTset | Malware, DoS, MitM, replay attacks |
| Lella et al. (2025) | Deep belief network (DBN) | CICIDS2017 | Brute-force, malware, phishing |
| Assiri & Ragab (2023) | Hybrid deep belief network (HDBN) | NSL-KDD | Various cyberattacks in NSL-KDD |
| Termos et al. (2024) | CNN, LSTM, GRU | Multiple (not named) | Multiple network attacks |
| Saurabh et al. (2024). | DT, KNN, LR, GNB & K-Means | CIC-ToN-IoT, KDD-99, NSL-KDD, CICIDS2017 | XSS, password, injection, scanning, backdoor, ransomware, MITM, DDoS, DoS |
| Haider et al. (2024) | XGBoost, Naive Bayes, AdaBoost, complement NB, GNB | UNSW-NB15 | DoS/DDoS, Sybil |
| Tyagi et al. (2024) | Deep learning, machine learning (Ensemble learning) | Various IoT security datasets (e.g., NSL-KDD, UNSW-NB15) | DoS, DDoS, Malware |
| Rasheed & Alnabhan (2024) | CatBoost, SVM, logistic regression | N-BaIoT dataset | Botnet, DDoS |
| Friha et al. (2023) | Deep neural networks (DNN), federated learning, differential privacy | IoT/IIoT dataset | Various IIoT-related attacks (DoS, command injection, etc.) |
| Sadhwani et al. (2025) | CNN, LSTM, Bi-LSTM (CNN-X best) | NSL-KDD, UNSW-NB15, TON-IoT, X-IIoTID | DoS, DDoS, injection, XSS, password, ransomware, backdoor, MITM |
| Saleh et al. (2025) | ANN | ToN-IoT telemetry | Password, scanning, XSS, DDoS, ransomware, injection, backdoor |
| Tawfik (2024) | Stacked autoencoders, CatBoost, transformer-CNN-LSTM ensemble | NSL-KDD, UNSW-NB15, AWID | DDoS, malware, anomaly attacks |
| Sneha & Prasad (2024) | ANN, CNN, LASSO | ToN-IoT | DDoS, general IIoT threats |
| Siganos et al. (2023) | SVM, RF, XGBoost, DNN, etc. | CIC-IoT-2022, IEC 60870-5-104 | MITM, DoS, unauthorized access, brute-force |
| Allafi & Alzahrani (2024) | BiLSTM, GRU, ELM | Benchmark IoT dataset | DDoS, jamming, DoS, flooding, botnet |
| Prasad et al. (2025) | Conditional variational autoencoder (CVAE), hybrid coati-grey wolf optimization | RT-IoT2022 dataset | Black-box, white-box, gray-box attacks |
| Zhang et al. (2024) | CNN, BiGRU, conditional denoising diffusion probabilistic model (CDDPM), SHAP | CICD-DOS2019, CICIDS2017 | DDoS, botnets, brute force, web attacks |
| Hamouda et al. (2024) | Conditional GAN (cGAN), federated learning (FL) | EdgeIIoTSet 2022 | DDoS, SQL injection, ransomware, MITM, etc. |
| Keshk et al. (2023) | Long short-term memory (LSTM) | NSL-KDD, UNSW-NB15, TON_IoT | Denial of service (DoS), Probe, U2R, R2L, DDoS, XSS, backdoor, worms, reconnaissance, and more |
| Konda, Ayyannan & Chandramouli (2023) | Random forest (RF), Naive Bayes (NB), XGBoost | Custom dataset of URLs (phishing and safe) | Phishing, DDoS |

| Authors name | AI models used | Datasets used | Threat scenarios (Types of attacks) |
|---|---|---|---|
| Termos et al. (2023) | Neural networks, decision tree, random forest, AdaBoost, XGBoost | IoT-NI dataset, Edge-IIoTset dataset | DDoS, DoS, MITM, injection attacks, malware |
| Ahanger (2018) | Artificial neural network (ANN) | Custom IoT network dataset (DDoS, Normal traffic) | DDoS, DoS, spoofing, device tampering, privacy breach |
| Shtayat et al. (2023) | CNN, ELM, ensemble learning | ToN_IoT | DoS, DDoS, injection, XSS, MITM, ransomware, password cracking, scanning, backdoor |
| Oseni et al. (2023) | Deep learning (SHAP) | ToN_IoT | DoS, DDoS, MITM, password cracking, ransomware, spoofing |
| Benaddi et al. (2022) | CNN, LSTM, GAN | Bot-IoT | DDoS, DoS, OS Scan, Keylogging, Data Exfiltration |
| Manivannan (2023) | Conjugate gradient-based improved GAN (CG-IGAN) | BoT-IoT | Malicious IoT data (botnet, malware, etc.) |
| Siddharthan, Deepa & Chandhar (2022) | Logistic regression, KNN, random forest, Naive Bayes, SVM, gradient boosting, decision tree | SENMQTT-SET (proposed), real testbed | DoS (Basic connect flooding), attack on broker/subscriber |
| Mousavi, Sadeghi & Sirjani (2023) | Logistic regression, random forest, k-NN, SVM, XGBoost | UNSW-NB15 | Multiple (from UNSW-NB15: DoS, exploits, etc.) |
| Almiani et al. (2020) | Deep recurrent neural network (RNN) | NSL-KDD (balanced version) | DoS, probe, R2L, U2R (from NSL-KDD) |
| Hasan, Moon & Raza (2023) | Ensemble learning (details N/A, LSTM, GRU in related work) | UNSW-NB (UNSW-NB15/ UNSW-NB18 referenced) | Nine attack families (Fuzzers, analysis, backdoors, DoS, exploits, surveillance, etc.) |
| Awotunde & Misra (2022) | Particle swarm optimization (PSO) + convolutional neural network (CNN) | CIC-IDS2017, UNSW-NB15 | Varied (CIC-IDS2017, UNSW-NB15 attacks) |
| Kim et al. (2024) | Artificial neural network (ANN) | ToN IoT telemetry | Password, scanning, XSS, DDOS, ransomware, injection, backdoor (ToN IoT) |
| Alzubi et al. (2025) | Decision tree | NFC-SECICIDS2018v2, BotIoT2018 | Anomalies and attacks in IoT |
| Serrano (2025) | LSTM, SVM | CIC-IDS-2018, BoT-IoT-2019, CIC-IoT-2023 | Multiple attack types |
| Arslan et al. (2024) | 1D CNN | Edge-IIoTset | Nine cyberattack types |
| Chandnani et al. (2025) | Federated multi-layered deep learning (Fed-MLDL) | CICIoT23, CICIoT22, ToN_IoT, Edge_IIoT, IoT-23 | DoS, DDoS, web spoofing |
| Imtiaz et al. (2025) | Convolutional neural networks (CNNs) | KDD CUP99, UNSW NB15, Bot-IoT | Botnets, DDoS, various cyber attacks |
| Ahmed et al. (2025) | XGBoost, LightGBM | RT-IoT2022 | Brute-force SSH, DDoS, Nmap scanning |
| Rehman et al. (2025) | Convolutional neural networks (CNN) | Edge-IIoTset, CIC-IDS2017 | Jamming, spoofing |
| Attique et al. (2024) | BiLSTM with self-adaptive attention | CICIDS2017, X-IIoTID | Various IIoT threats |
| Kantharaju et al. (2024) | SAPGAN (Self-attention progressive GAN) | Bot-IoT | DDoS, flood attacks, RTSP brute force |
| Bar, Prasad & Sayeed (2024) | Graph neural networks (GNNs), federated learning | N/A (Review study) | DDoS, label flipping, MiTM, zero-day exploits |
| Ben Atitallah et al. (2024) | Deep infomax (DIM), prototypical networks, random forest | MaleVis, WSN-DS | Malware detection in IoT |
| Alrayes et al. (2024) | Denoising autoencoder (DAE), GRU, LSTM ensemble | Benchmark database (not specified) | DDoS attacks |

(Continued)

| Table 5 (continued) | | | |
|---|---|---|---|
| Authors name | AI models used | Datasets used | Threat scenarios (Types of attacks) |
| *Behera et al. (2024)* | CNN, Bi-GRU, Bi-LSTM (hybrid DNN) | InSDN, UNSW-NB15, CICIoT2023 | DDoS, probe, ransomware, botnet |
| *Jouhari & Guizani (2024)* | CNN-BiLSTM | UNSW-NB15 | Various IoT attacks |
| *Chelghoum et al. (2024)* | Deep learning (CNIDS approach) | Proof of concept (Dataset not explicitly mentioned) | Zero-day attacks, falsified attacks |
| *El-Shafeiy et al. (2024)* | CNN, complex gated recurrent networks (CGRNs) | UNSW-NB15, KDDCup99, IoT-23 | Sophisticated cyber-attacks in IoT |
| *Indra et al. (2024)* | Gradient boosting, random forest (Ensemble) | CICIDS2017 | Botnet, ransomware, jamming, backdoor, DDoS |

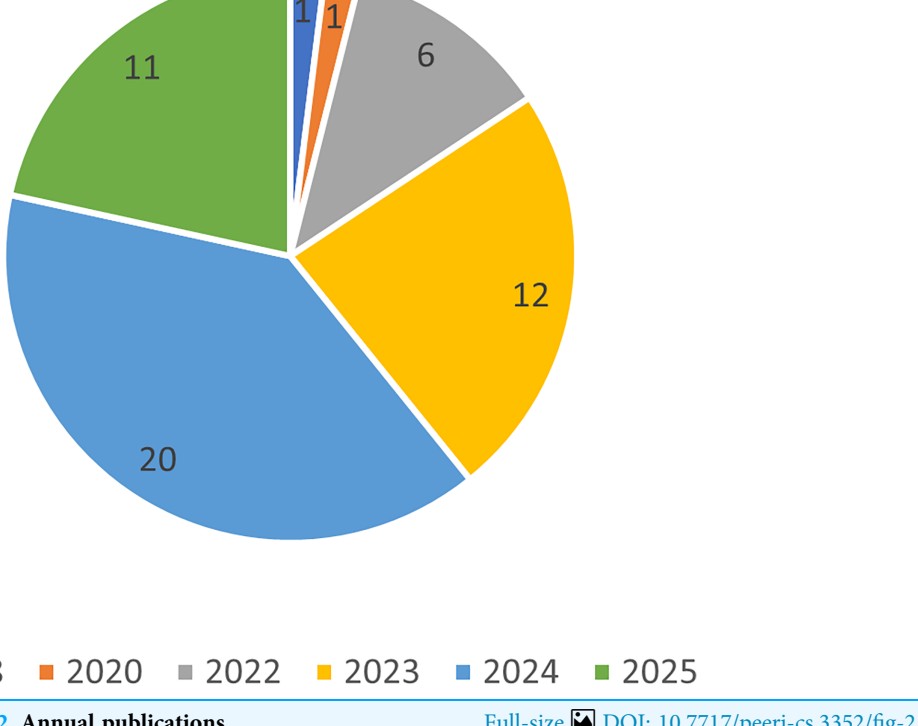

**Figure 2 Annual publications.**

robust national investments and access to extensive empirical data from major smart infrastructure initiatives. Research contributions from other regions were relatively sparse, indicating the existing global distribution of scholarly activity rather than any selection bias in our review process.
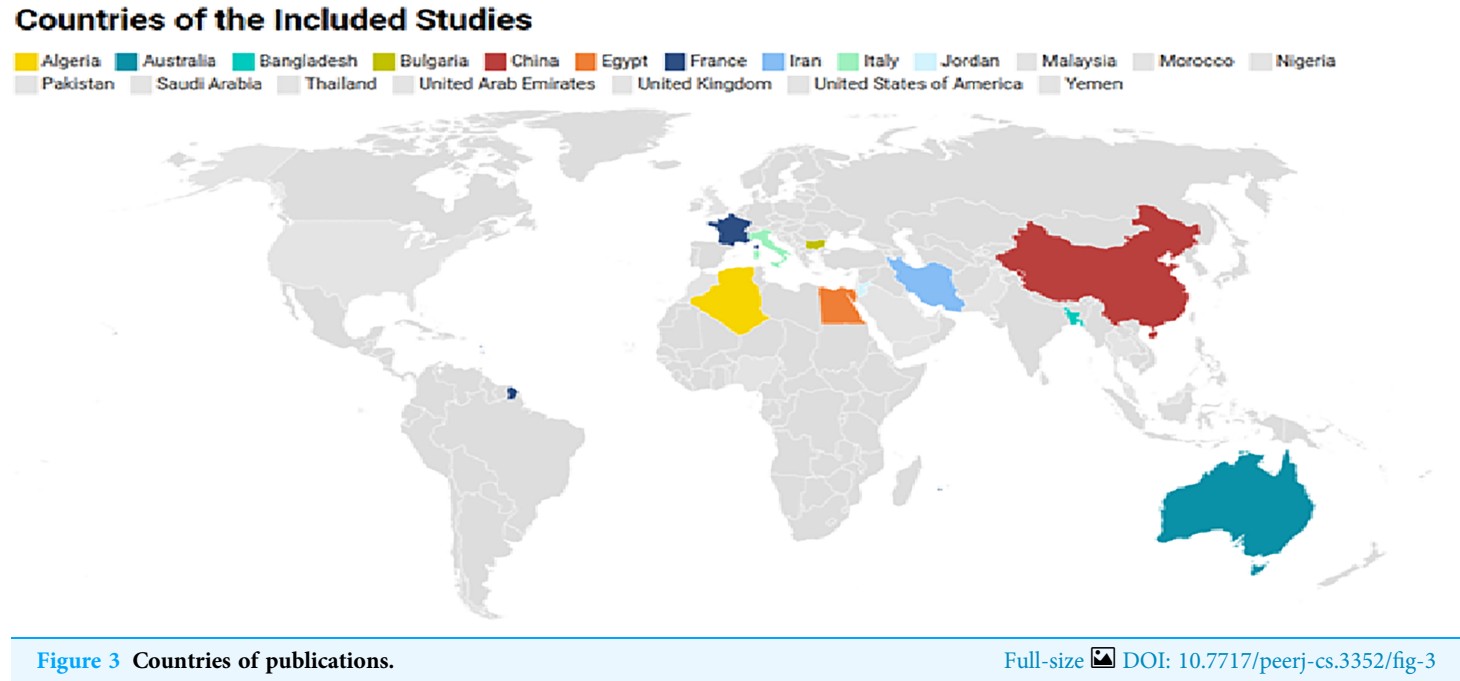

**Figure 3 Countries of publications.**

## Comparative analysis of AI models' performance metrics in IoT-based intrusion detection

As reported in Table 6, a broad range of AI models including machine learning, deep learning, and hybrid approaches have been applied to intrusion detection in IoT environments, exhibiting significant variation in their reported performance metrics. While some studies provided comprehensive evaluations including accuracy, precision, and F1-score, others omitted one or more of these key indicators. For instance, *Saheed, Omole & Sabit (2025)* and *Rasheed & Alnabhan (2024)* demonstrated near-perfect accuracy, precision, and F1 scores using models like long short-term memory (LSTM) and CatBoost. Some studies like *Gueye et al. (2023)*, *Sneha & Prasad (2024)* and *Kulrujiphat & Kulrujiphat (2024)* reported only general high accuracy without specifics, while others like *Tyagi et al. (2024)* and *Almiani et al. (2020)* did not provide any quantitative performance metrics. Notably, *Keshk et al. (2023)* and *Rehman et al. (2025)* were among the few that provided a comprehensive set of metrics across multiple datasets. These variations highlight the challenge of direct comparison but also underscore a consistent trend of AI models achieving strong performance in smart grid intrusion detection, even when one or more metrics were not reported. The overall trend highlights the growing efficacy of AI-driven systems, particularly ensemble and hybrid models, in addressing complex cybersecurity threats within IoT networks.

## Publication bias

The contour-enhanced funnel plot in Fig. 4 provides a visual assessment of potential publication bias in the meta-analysis evaluating the effectiveness of AI-driven IDS in IoT environments. Created using R version 4.4.3, the plot displays individual studies by their

**Table 6 Compartive analysis of AI performance.**

| Sources | AI models used | Reported accuracy | Reported precision | Reported F1 score |
|---|---|---|---|---|
| Gueye et al. (2023) | MLP, CNN, RNN with embedding layer | High (specific % not stated) | N/A | N/A |
| Saheed, Omole & Sabit (2025) | LSTM, attention mechanism | 99.98% (SWaT), 99.87% (WADI) | 99.98% (SWaT), 99.87% (WADI) | 99.98% (SWaT), 99.87% (WADI) |
| Kulrujiphat & Kulrujiphat (2024) | Various (including ML and DL models) | Varied across models | N/A | N/A |
| Lella et al. (2025) | Deep belief network (DBN) | 98.90% | N/A | N/A |
| Assiri & Ragab (2023) | Hybrid deep belief network (HDBN) | 99.21% | N/A | N/A |
| Termos et al. (2024) | CNN, LSTM, GRU | ~+7.7% | N/A | N/A |
| Saurabh et al. (2024) | DT, KNN, LR, GNB & K-means | 99.49% (known), 98.936% (unknown) | N/A | 98.43% to 99.49% |
| Haider et al. (2024) | XGBoost, Naive Bayes, AdaBoost, complement NB, GNB | Highest with AdaBoost | N/A | N/A |
| Tyagi et al. (2024) | Deep learning, machine learning (Ensemble learning) | N/A | N/A | N/A |
| Rasheed & Alnabhan (2024) | CatBoost, SVM, logistic regression | 92.98% (Logistic Regression), 93.27% (SVM), 99.49% (CatBoost) | 94.13% (SVM), 92.91% (Logistic Regression), 99.49% (CatBoost) | 91% (Deep Residue CNN) |
| Friha et al. (2023) | Deep neural networks (DNN), federated learning, differential privacy | 94.37% (compared to centralized approach) | 12% improvement compared to FL-based IDS solutions. | 9% improvement compared to FL-based IDS solutions. |
| Sadhwani et al. (2025) | CNN, LSTM, Bi-LSTM (CNN-X best) | 98.21% (NSL-KDD), 97.80% (TON-IoT), etc. | Not specified | 98.09% (X-IIoTID), etc. |
| Saleh et al. (2025) | ANN | 58–91% (varies by device) | Up to 100% | Up to 92% |
| Tawfik (2024) | Stacked autoencoders, CatBoost, transformer-CNN-LSTM ensemble | Over 99% | Not specified | Not specified |
| Sneha & Prasad (2024) | ANN, CNN, LASSO | Not numerically specified | High | High |
| Siganos et al. (2023) | SVM, RF, XGBoost, DNN, etc. | >99% (in some models) | Varies | Up to 99% |
| Allafi & Alzahrani (2024) | BiLSTM, GRU, ELM | 99.31% | N/A | N/A |
| Prasad et al. (2025) | Conditional variational autoencoder (CVAE), hybrid coati-grey wolf optimization | 99.91% | N/A | N/A |
| Zhang et al. (2024) | CNN, BiGRU, conditional denoising diffusion probabilistic model (CDDPM), SHAP | Not specified | Not specified | Not specified |
| Hamouda et al. (2024) | Conditional GAN (cGAN), federated learning (FL) | 92.72% (without DP) | N/A | N/A |
| Keshk et al. (2023) | Long short-term memory (LSTM) | NSL-KDD: 81.1%, UNSW-NB15: 86.6%, ToN_IoT: 87.3% | NSL-KDD: 92.1%, UNSW-NB15: 81.1%, ToN_IoT: 78.4% | NSL-KDD: 81.5%, UNSW-NB15: 89.1%, ToN_IoT: 82.9% |
| Konda, Ayyannan & Chandramouli (2023) | Random forest (RF), Naive Bayes (NB), XGBoost | 96.98 | 95.46 | 95.75 |
| Termos et al. (2023) | Neural networks, decision tree, random forest, AdaBoost, XGBoost | 95.5 | N/A | N/A |
| Ahanger (2018) | Artificial neural network (ANN) | >99% | N/A | N/A |

| Sources | AI models used | Reported accuracy | Reported precision | Reported F1 score |
|---|---|---|---|---|
| *Shtayat et al. (2023)* | CNN, ELM, ensemble learning | 99.15% | N/A | 98.83% |
| *Oseni et al. (2023)* | Deep learning (SHAP) | 99.15% | N/A | 98.83% |
| *Benaddi et al. (2022)* | CNN, LSTM, GAN | 40% increase for Theft Attacks | N/A | N/A |
| *Manivannan (2023)* | Conjugate gradient-based improved GAN (CG-IGAN) | 99.10% | 97.25% | N/A |
| *Siddharthan, Deepa & Chandhar (2022)* | Logistic regression, KNN, random forest, Naive Bayes, SVM, gradient boosting, decision tree | >99% | N/A | N/A |
| *Mousavi, Sadeghi & Sirjani (2023)* | Logistic regression, random forest, k-NN, SVM, XGBoost | XGBoost highest (exact value N/A) | N/A | N/A |
| *Almiani et al. (2020)* | Deep recurrent neural network (RNN) | N/A | N/A | N/A |
| *Hasan, Moon & Raza (2023)* | Ensemble learning (details N/A, LSTM, GRU in related work) | 97.68% | N/A | N/A |
| *Awotunde & Misra (2022)* | Particle swarm optimization (PSO) + Convolutional neural network (CNN) | 99.45% | N/A | N/A |
| *Saleh et al. (2025)* | ANN | 91–100% | 91–100% | 91–100% |
| *Alzubi et al. (2025)* | Decision tree | 97.59% (NFC-SECICIDS2018v2), 99.97% (BotIoT2018) | N/A | N/A |
| *Serrano (2025)* | LSTM, SVM | N/A (relative comparison: LSTM better than SVM by 30%) | N/A | N/A |
| *Arslan et al. (2024)* | 1D CNN | 99.90% | N/A | N/A |
| *Chandnani et al. (2025)* | Federated multi-layered deep learning (Fed-MLDL) | 98.1–99.2% | N/A | N/A |
| *Imtiaz et al. (2025)* | Convolutional neural networks (CNNs) | 99.34% (KDD CUP99), 99.61% (UNSW NB15), 99.21% (Bot-IoT) | N/A | N/A |
| *Ahmed et al. (2025)* | XGBoost, LightGBM | 99.553% (XGBoost), 99.651% (LightGBM) | N/A | N/A |
| *Rehman et al. (2025)* | CNN | 93.4% (Edge-IIoTset), 95.8% (CIC-IDS2017) | 88% (Edge-IIoTset), 94.9% (CIC-IDS2017) | 87% (Edge-IIoTset), 93% (CIC-IDS2017) |
| *Attique et al. (2024)* | BiLSTM with self-adaptive attention | 99.92% (CICIDS2017), 96.54% (X-IIoTID) | N/A | N/A |
| *Kantharaju et al. (2024)* | SAPGAN (Self-attention progressive GAN) | 23.19%–27.55% higher than baselines | N/A | Higher than baselines |
| *Bar, Prasad & Sayeed (2024)* | Graph neural networks (GNNs), federated learning | >99% (GNN models) | N/A | N/A |
| *Ben Atitallah et al. (2024)* | Deep infomax (DIM), prototypical networks, random forest | 98.60% (MaleVis), 99.56% (WSN-DS) | 98.79% (MaleVis), 99.56% (WSN-DS) | 98.63% (MaleVis), 99.56% (WSN-DS) |
| *Alrayes et al. (2024)* | Denoising autoencoder (DAE), GRU, LSTM ensemble | Higher than comparative DL techniques | N/A | N/A |
| *Behera et al. (2024)* | CNN, Bi-GRU, Bi-LSTM (hybrid DNN) | Very high, outperforming baselines | N/A | N/A |
| *Jouhari & Guizani (2024)* | CNN-BiLSTM | 97.28% (binary), 96.91% (multi-class) | N/A | N/A |

(Continued)

| Table 6 (continued) | | | | |
| --- | --- | --- | --- | --- |
| Sources | AI models used | Reported accuracy | Reported precision | Reported F1 score |
| *Chelghoum et al. (2024)* | Deep learning (CNIDS approach) | High accuracy, validated by simulation | N/A | N/A |
| *El-Shafeiy et al. (2024)* | CNN, complex gated recurrent networks (CGRNs) | 99.20% | N/A | N/A |
| *Indra et al. (2024)* | Gradient boosting, random forest (Ensemble) | 98.75% | 98.70% | 96.90% |

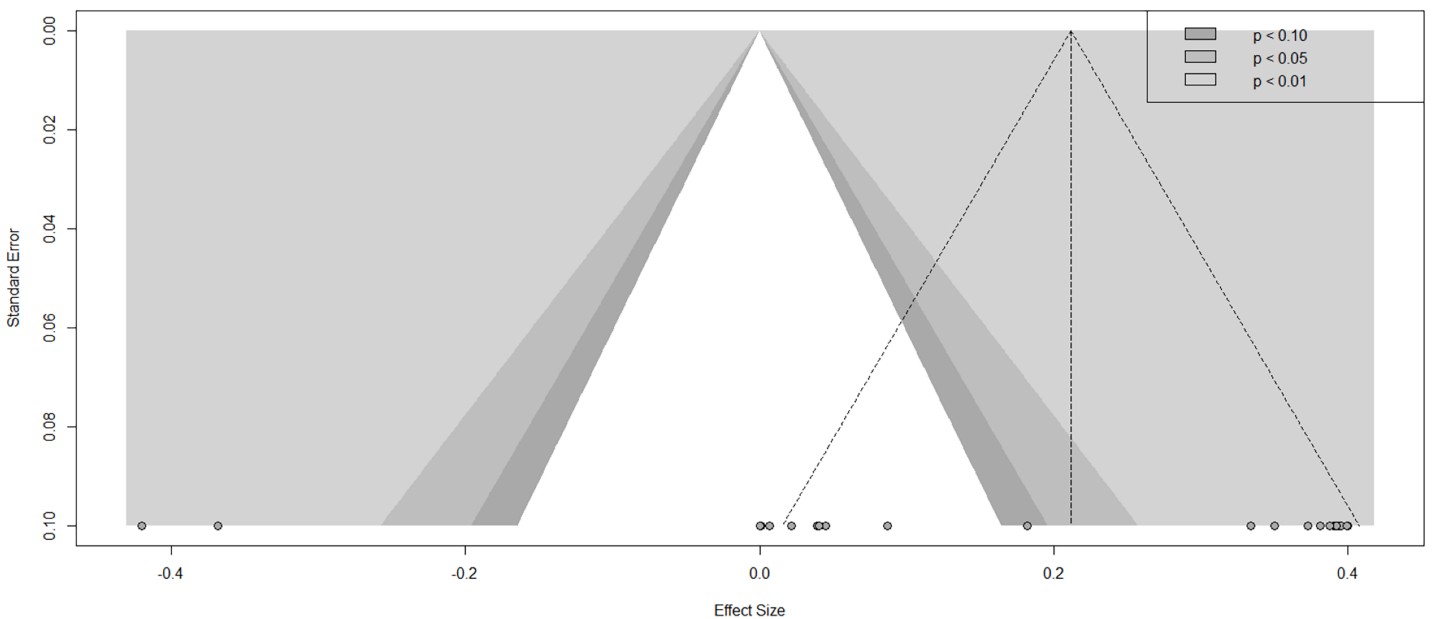

**Figure 4 Funnel plot showing publication bias.**

effect sizes (x-axis) and standard errors (y-axis), where studies with greater precision (smaller standard errors) are positioned higher, and those with less precision appear lower. In a scenario free from publication bias data points would be expected to distribute symmetrically around the central vertical line representing the pooled effect estimate forming a characteristic inverted funnel. In this case, while the plot retains a rough funnel shape a clear asymmetry is evident particularly a concentration of studies on the right side and a relative absence on the left especially in zones indicating statistically significant negative effects. This skew implies a potential bias toward publishing studies with favorable or positive outcomes for AI-based IDS models, while those reporting null or negative effects appear underrepresented. The shaded regions within the funnel reflect different levels of statistical significance with lighter zones indicating highly significant findings. A substantial number of studies fall within or close to these regions, further suggesting a tendency toward selective reporting of significant results. Such asymmetry may also be partially attributed to genuine heterogeneity stemming from differences in AI model types, datasets, or evaluation methodologies rather than publication bias alone.

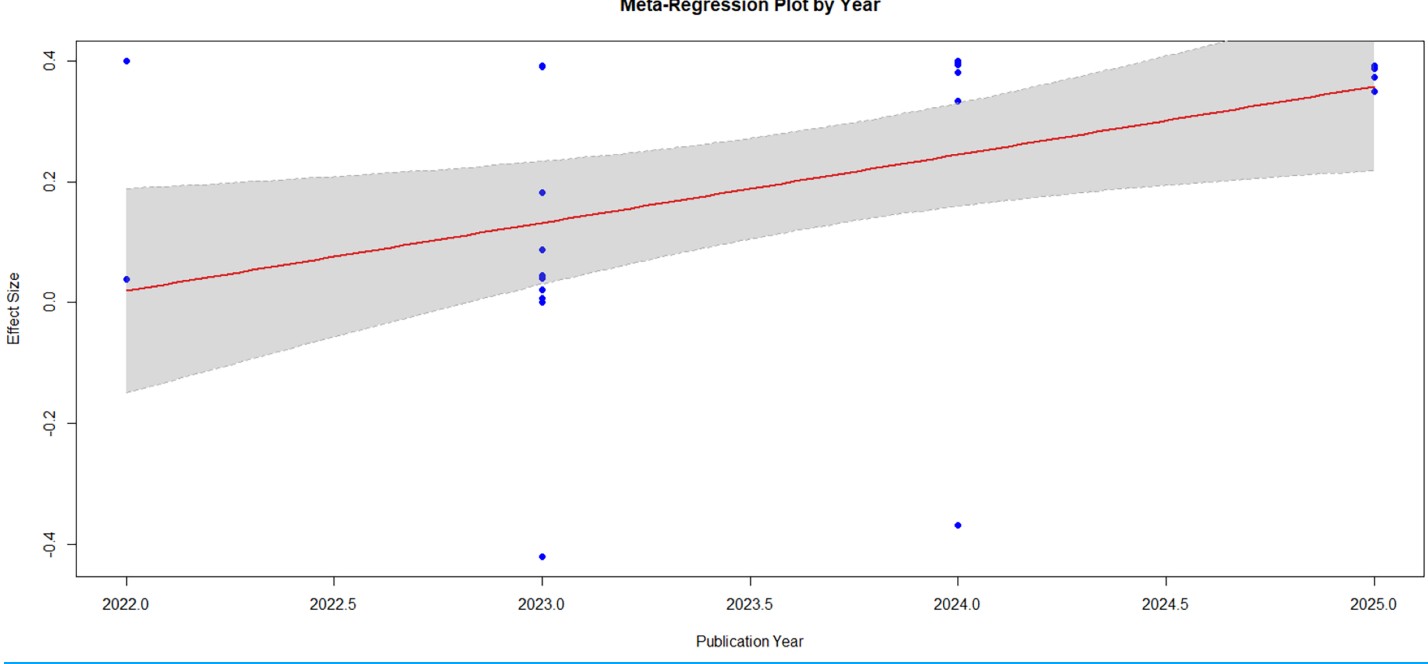

**Figure 5** Regression plot showing the publication years. 

The scarcity of studies in the top-left or far-left portions of the plot representing strong negative or non-significant results raises concern about potential underreporting. This has important implications, as unaddressed publication bias can lead to inflated pooled effect size estimates and overly optimistic conclusions. While contour-enhanced funnel plots improve interpretability over traditional plots by incorporating significance thresholds they remain primarily qualitative tools. In this study, Egger's regression test confirmed statistically significant asymmetry ($p < 0.001$) providing quantitative evidence of potential publication bias in the included literature.

## Meta-regression plot showing the publication years

The meta-regression plot generated using R version 4.4.3 provides valuable insights into how the effect sizes of studies evaluating AI-driven IDS in IoT environments have evolved over time as shown in Fig. 5. This plot, which regresses effect sizes on publication years, reveals a discernible upward trend, suggesting that more recent studies report stronger performance advantages for AI-based IDS solutions compared to traditional techniques. The red regression line indicates a positive linear relationship, meaning that as the years progress from 2022 to 2025, the reported effect sizes increase. This trend is visually supported by the spread of data points, with earlier studies generally showing lower or even negative effect sizes, while later studies tend to cluster above the zero baseline, indicating better outcomes in favor of AI. The gray shaded region surrounding the regression line represents the 95% confidence interval, which quantifies the uncertainty around the predicted effect sizes. While the interval is relatively narrow for mid-range years like 2023, it widens towards 2025, which may be attributed to a smaller number of studies in the

more recent publication years or increased variability in effect size reporting. This widening band suggests that although the general trend is upward, there is some variability in how recent studies assess and report AI effectiveness, possibly due to differences in methodology, datasets, evaluation metrics, or model complexity.

The positive association between effect size and publication year may reflect genuine progress in AI technologies. In the context of IoT security, newer AI models such as deep neural networks, convolutional and recurrent architectures, and hybrid ensemble methods have been designed to tackle increasingly complex and high-dimensional data characteristic of IoT traffic. Advances in training algorithms, model optimization, and real-time processing capabilities have also contributed to more robust and adaptive intrusion detection systems. Consequently, newer studies are more likely to showcase these advancements, resulting in higher reported performance metrics compared to earlier work that relied on more conventional or rule-based approaches. This pattern may reflect a growing emphasis in the research community on benchmarking AI solutions against rigorous and diverse datasets. As the field matures, researchers are adopting more standardized evaluation frameworks and are likely to report not just accuracy, but also precision, recall, and F1-scores, providing a more holistic view of model performance. This evolution in research practices may partially explain the improved effect sizes in recent studies. The increase in computational resources and the accessibility of open-source AI libraries have lowered the barrier for deploying state-of-the-art techniques, allowing more comprehensive experimentation and evaluation in academic studies.

## Artificial intelligence techniques used

The findings derived from this meta-analysis based on AI techniques applied in IDSs for IoT ecosystems predominantly fall under two overarching categories: machine learning and deep learning. Within traditional machine learning, methods such as SVM, RF, decision trees (DT), k-nearest neighbors (k-NN), naïve Bayes (NB), and gradient boosting are frequently employed (*Allafi & Alzahrani, 2024*; *Alzubi et al., 2025*; *Assiri & Ragab, 2023*; *Ben Atitallah et al., 2024*; *Benaddi et al., 2022*; *Friha et al., 2023*; *Haider et al., 2024*; *Hamouda et al., 2024*; *Indra et al., 2024*; *Kantharaju et al., 2024*; *Lella et al., 2025*; *Mousavi, Sadeghi & Sirjani, 2023*; *Sadhwani et al., 2025*; *Saurabh et al., 2024*; *Sejaphala, Malele & Lugayizi, 2024*; *Shtayat et al., 2023*; *Termos et al., 2023*). On the other hand, deep learning approaches commonly involve CNNs, RNNs, LSTM networks, and Autoencoders (*Oseni et al., 2023*). These approaches are favored across a large number of reviewed studies for their capacity to process extensive, high-dimensional data and to adapt dynamically to emerging cyberattack patterns (*Allafi & Alzahrani, 2024*; *Assiri & Ragab, 2023*; *Chandnani et al., 2025*; *Gueye et al., 2023*; *Imtiaz et al., 2025*; *Keshk et al., 2023*; *Shtayat et al., 2023*; *Siddharthan, Deepa & Chandhar, 2022*; *Sneha & Prasad, 2024*; *Termos et al., 2024*).

A notable trend in recent literature is the preference for hybrid methodologies that integrate multiple AI algorithms. This includes ensemble approaches that combine several classifiers to mitigate bias and variance, as well as layered system architectures where each stage utilizes a distinct model to enhance detection accuracy (*Bar, Prasad & Sayeed, 2024*; *Mallidi & Ramisetty, 2025*; *Mishra & Pandya, 2021*; *Salem et al., 2024*; *Serrano, 2025*;

_Siganos et al., 2023_). For example, one layer might utilize random forests for initial anomaly detection, followed by deep learning models for advanced classification. These integrated systems often demonstrate improved performance across diverse evaluation metrics, as they capitalize on the unique strengths of each algorithm (_Bar, Prasad & Sayeed, 2024_; _Manivannan, 2023_; _Shtayat et al., 2023_; _Tawfik, 2024_). There has been a modest but growing interest in explainable AI (XAI) techniques within the IDS (_Attique et al., 2024_; _Keshk et al., 2023_; _Lella et al., 2025_; _Sadhwani et al., 2025_). Although most research continues to focus primarily on optimizing quantitative metrics like accuracy and precision, a subset of studies has begun to explore model interpretability to enhance transparency and trustworthiness. This movement reflects an emerging recognition that model decisions must be explainable to cybersecurity professionals and system stakeholders, especially in critical infrastructure settings (_Lella et al., 2025_; _Rehman et al., 2025_; _Saheed, Omole & Sabit, 2025_; _Saleh et al., 2025_).

## Meta-analysis results by metric

This meta-analysis consolidates findings from selected studies to assess the comparative effectiveness of AI-driven IDS _vs_ traditional methods across various IoT application domains. Traditional IDS typically depend on rule-based or signature-based mechanisms, relying on fixed attack patterns and expert-defined rules for anomaly detection. In contrast, AI-based IDS utilize machine learning and deep learning models that support adaptive data-driven detection of new and evolving threats. The aggregated evidence demonstrates a clear and statistically significant advantage in performance for AI-based approaches. Both fixed and random effects models yielded a mean accuracy difference (MD) of 0.2115, suggesting that AI-driven systems outperformed traditional IDS by an average of 21.15% in accuracy (_Ahmed et al., 2025_; _Behera et al., 2024_; _Prasad et al., 2025_; _Tyagi et al., 2024_). This performance gain was statistically robust ($p < 0.0001$), with 95% confidence intervals of [0.1738, 0.2492] under the fixed-effects model and [0.1176, 0.3054] under the random-effects model, corroborating earlier research supporting the superiority of AI-based IDS (_Konda, Ayyannan & Chandramouli, 2023_; _Kulrujiphat & Kulrujiphat, 2024_; _Lella et al., 2025_; _Rasheed & Alnabhan, 2024_). Substantial heterogeneity was observed across studies ($I^2 = 82.3\%$; Q = 146.53, df = 26, $p < 0.0001$; $\tau^2 = 0.0464$), likely due to variations in datasets, algorithmic models, and evaluation protocols.

However, relying on accuracy alone can be misleading particularly with imbalanced datasets common in IDS benchmarks such as NSL-KDD, CICIDS2017, ToN-IoT, BoT-IoT, and Edge-IIoTset. High accuracy may mask poor performance on minority (malicious) classes. Precision, which measures the proportion of true positives among predicted positives reflects a model's ability to avoid false alarms metrics where algorithms like SVM and RF tend to perform well (_Abdullahi et al., 2022_; _Lella et al., 2025_; _Rehman et al., 2025_). Recall indicating how effectively a model identifies actual threats is especially important in high-risk environments and is typically higher in deep learning models such as RNNs and LSTMs (_Bar, Prasad & Sayeed, 2024_; _Mousavi, Sadeghi & Sirjani, 2023_; _Rehman et al., 2025_; _Saheed, Omole & Sabit, 2025_). The F1-score balances precision and recall offering a comprehensive view of detection performance. Hybrid and ensemble

models often excel in this metric by better managing trade-offs between sensitivity and specificity. Other performance indicators including AUC-ROC, Matthews correlation coefficient (MCC), and detection latency are also employed to account for class imbalance and real-time requirements in IoT scenarios (*Imtiaz et al., 2025*; *Mishra & Pandya, 2021*; *Sneha & Prasad, 2024*).

## DISCUSSION

The results confirm a rapid increase in AI-driven IDS research especially after 2022 reflecting the global urgency to secure IoT and smart grid infrastructures. India, Saudi Arabia, and China dominate scholarly output, largely due to strong national investments in cybersecurity and smart grid systems. Performance comparisons highlight that while machine learning models provide baseline detection, deep learning and hybrid approaches yield superior outcomes particularly in recall and F1-scores. The growing trend toward hybrid systems indicates an effort to combine the strengths of diverse algorithms for robust detection. The study shows the superiority of AI-based IDS over traditional rule-based systems though the presence of publication bias suggests results should be interpreted cautiously. The upward trend in effect sizes over time likely reflects advances in computational capabilities, dataset diversity, and methodological rigor. The emerging adoption of explainable AI signals a shift toward not only accuracy but also interpretability critical for real-world cybersecurity applications.

## CONCLUSIONS

This meta-analysis offers a well-rounded look at how effective AI-driven IDSs are in protecting IoT networks. By analyzing findings from numerous empirical studies, it becomes clear that AI technologies especially those that use machine learning and deep learning consistently outperform traditional intrusion detection systems when it comes to identifying and managing cyber threats. Across core performance indicators like accuracy, precision, recall, and F1-score, AI-based methods show a clear advantage. While individual AI models tend to perform better on some metrics than others, hybrid or ensemble approaches where multiple models are combined typically offer the most balanced and robust results. This suggests that using a mix of AI strategies can be particularly effective in dealing with the complexity and unpredictability of IoT environments. The analysis also found that performance has generally improved over time. This likely reflects progress in computing capabilities, the availability of more comprehensive training data, and advancements in algorithms. However, the results also point to some issues like potential publication bias and high variability among studies. These factors serve as a reminder that results should be interpreted with an understanding of their specific context. This study supports the growing belief that AI can play a transformative role in cybersecurity especially in IoT networks, which are often resource-limited and highly interconnected. The quantitative evidence gathered here strengthens the case for integrating AI more confidently into practical cybersecurity solutions.

## Theoretical contributions

This meta-analytical review makes several important theoretical contributions to the growing field of AI in IDS for IoT environments. First and foremost, the study consolidates fragmented empirical findings and offers a structured overview of how different AI techniques spanning machine learning, deep learning, hybrid models, and ensemble methods perform across key evaluation metrics. This synthesis not only highlights performance trends but also contributes to theory-building by identifying patterns and relationships that are not easily observable through individual studies. By examining metrics like accuracy, precision, recall, and F1-score in tandem, the review strengthens the theoretical understanding of trade-offs between detection capabilities and error rates, which are critical when designing IDS models. Second, the study extends the theoretical discourse on model selection and optimization in cybersecurity applications. It suggests that no single AI model universally excels across all performance indicators. Rather, context-specific choices such as whether to prioritize minimizing false positives (precision) or maximizing threat detection (recall) must be informed by the specific demands and vulnerabilities of the target system. This nuanced insight adds depth to existing models of AI application in security, encouraging a more layered and adaptive approach. Third, the inclusion of meta-regression analysis to assess changes in AI-IDS performance over time adds a temporal dimension to theoretical discussions. It reflects how improvements in computational capacity, algorithmic development, and dataset quality influence outcomes thus integrating technological evolution into the theoretical framework of cybersecurity research.

## Practical implications

The findings of this meta-analysis hold meaningful practical implications for cybersecurity professionals, IoT system designers, policymakers, and organizations deploying AI-based IDSs. Firstly, the consistent outperformance of AI-IDSs across key metrics especially accuracy, precision, recall, and F1-score provide strong evidence for practitioners to consider transitioning from traditional IDS solutions to AI-driven ones. These results suggest that AI models are better equipped to handle the complexity and volume of threats in modern IoT environments, which often involve vast numbers of connected devices generating high-frequency, heterogeneous data. Secondly, the variation in performance across different AI techniques offers practical guidance for model selection. For instance, organizations that prioritize minimizing false alarms may lean toward machine learning models like random forest or support vector machines, which showed higher precision. On the other hand, sectors with a low tolerance for missed threats such as healthcare or critical infrastructure may benefit more from deep learning models like RNNs or CNNs, which offer higher recall. The study also underscores the advantages of using hybrid or ensemble methods to strike a balance between these competing demands, making them ideal for complex or high-stakes environments. Third, the upward trend in performance over time highlighted by the meta-regression analysis reflects real-world advancements in AI capability and data accessibility. This evolution provides reassurance to stakeholders that AI-IDSs are not static technologies but are continually improving. However, it also signals

the need for ongoing investment in skills development, infrastructure upgrades, and periodic reassessment of deployed models to ensure they remain effective. The detection of publication bias and performance inflation in smaller studies serves as a cautionary note. It encourages organizations to critically evaluate vendor claims or academic benchmarks and to conduct in-house testing where possible before full-scale deployment. Finally, the study advocates for the standardization of evaluation protocols and reporting practices, which would significantly improve practical decision-making. Clearer benchmarks would allow practitioners to compare tools more easily, assess cost-benefit trade-offs, and justify investments to stakeholders. There is interest in XAI Of the 51 studies reviewed six (*Attique et al., 2024*; *Keshk et al., 2023*; *Sadhwani et al., 2025*; *Shtayat et al., 2023*; *Siganos et al., 2023*; *Sneha & Prasad, 2024*) (approximately 11.8%) specifically focused on XAI, indicating a growing yet still limited emphasis on model interpretability in the context of IoT intrusion detection research.

Although the meta-analysis demonstrates the superior performance of AI-driven IDS models, their implementation in real-world IoT environments faces several practical obstacles. One major issue is latency, as many IoT security scenarios demand real-time or near-real-time detection to prevent the rapid spread of attacks. Another challenge is the limited energy capacity of battery-operated or resource-constrained IoT devices, which restricts the use of computationally heavy models like those based on deep learning. Additionally, hardware limitations must be taken into account, since many edge and embedded devices lack the necessary processing power and memory to support complex AI algorithms efficiently. To overcome these challenges, strategies such as lightweight model optimization, hardware acceleration (*e.g.*, using specialized edge AI chips), and energy-efficient system design will be essential for the practical and sustainable deployment of IDS solutions.

## Limitations and future directions

While this meta-analysis offers valuable insights into the effectiveness of AI-driven IDSs in IoT environments, it is important to acknowledge several limitations that also point to promising directions for future research. One key limitation is the heterogeneity among the included studies. Differences in datasets, evaluation metrics, and implementation contexts introduced variability that complicates direct comparisons and may reduce the precision of aggregated results. To enhance comparability and replicability, future research should prioritize the use of standardized benchmarking practices and publicly available, diverse IoT datasets. Publication bias presents another concern, as indicated by funnel plot asymmetry. The tendency to publish only positive or statistically significant findings may inflate perceptions of AI-IDS efficacy. To address this, future reviews should actively incorporate grey literature such as technical reports, dissertations, and conference articles to build a more comprehensive and unbiased evidence base. A further limitation is the predominance of simulation-based studies. Most AI-IDS models were evaluated using static datasets in offline environments, which may not accurately reflect the challenges of real-world IoT systems characterized by dynamic data, limited computational resources, and unpredictable network conditions. Future studies should emphasize real-time

deployment and performance validation in operational IoT settings. Interpretability remains a pressing issue. While this study focused primarily on accuracy-related performance metrics, the lack of attention to model transparency can hinder adoption in security-sensitive environments. Future research should investigate how XAI techniques influence user trust, system accountability, and decision-making efficacy in cybersecurity contexts. With the rapid evolution of cyber threats including adversarial machine learning, zero-day exploits, and domain-specific vulnerabilities there is a clear need for adaptive AI-IDS models capable of responding to novel attack vectors. Continuous learning and regular model updates will be essential to ensure lasting effectiveness. The review reveals a geographic skew toward studies conducted in India, Saudi Arabia, and China, reflecting both the global distribution of publications and the accessibility of data in these fields. The limited representation from regions such as Africa, South America, and certain parts of Europe constrains the broader applicability of the findings. To enhance global relevance, future research should emphasize cross-regional studies that capture a more diverse range of contexts.

## ABBREVIATIONS

| | |
|---|---|
| ADA | AdaBoost |
| AI | artificial intelligence |
| ANN | artificial neural network |
| AUC-ROC | area under the receiver operating characteristic curve |
| AWID | Aegean Wi-Fi Intrusion Dataset |
| BiGRU | bidirectional gated recurrent unit |
| BiLSTM/Bi-LSTM | bidirectional long short-term memory |
| CatBoost | categorical gradient boosting |
| CDDPM | conditional denoising diffusion probabilistic model |
| cGAN | conditional generative adversarial network |
| CGRN | complex gated recurrent network |
| CIC-IDS2017 | Canadian Institute for Cybersecurity Intrusion Detection 2017 dataset |
| CIC-IoT-2022/CICIoT2023 | Canadian Institute for Cybersecurity IoT datasets (2022/2023) |
| CICD-DOS2019 | Canadian Institute for Cybersecurity DoS 2019 dataset |
| CNN | convolutional neural network |
| CVAE | conditional variational autoencoder |
| DAE | denoising autoencoder |
| DBN | deep belief network |
| DDoS | distributed denial-of-service |
| DL | deep learning |
| DNN | deep neural network |
| DoS | denial-of-service |
| DP | differential privacy |

| | |
|---|---|
| DT | decision tree |
| ELM | extreme learning machine |
| FL | federated learning |
| GAN | generative adversarial network |
| GNB | Gaussian naïve Bayes |
| GRU | gated recurrent unit |
| HDBN | hybrid deep belief network |
| IDS | intrusion detection system |
| IEC 60870-5-104 | telecontrol protocol for industrial systems |
| IIoT | Industrial Internet of Things |
| InSDN | intrusion dataset for software-defined networking |
| IoT | Internet of Things |
| KDD-99/KDDCup99 | Knowledge Discovery in Databases 1999 dataset |
| KNN | k-nearest neighbors |
| LASSO | least absolute shrinkage and selection operator |
| LR | logistic regression |
| LSTM | long short-term memory |
| MCC | Matthews correlation coefficient |
| MITM | man-in-the-middle |
| NB | naïve Bayes |
| NSL-KDD | refined version of KDD-99 dataset |
| PRISMA | Preferred Reporting Items for Systematic Reviews and Meta-Analyses |
| RF | random forest |
| RNN | recurrent neural network |
| RT-IoT2022/X-IIoTID | real-time/extended IIoT intrusion datasets (as cited in tables) |
| SHAP | SHapley Additive exPlanations |
| SQL | Structured Query Language |
| SVM | support vector machine |
| ToN-IoT/TON_IoT | Telemetry/Traces of Networked IoT datasets |
| UNSW-NB15 | University of New South Wales network-based 2015 dataset |
| U2R | user-to-root (attack class) |
| R2L | remote-to-local (attack class) |
| XGBoost | Extreme Gradient Boosting |
| XSS | cross-site scripting |

## ACKNOWLEDGEMENTS

OpenAI's ChatGPT (GPT-4) was used for grammar checking and for verification of extracted data against the original article.

### Funding

This work was supported by Hunan Key Laboratory for Internet of Things in Electricity. The funders had no role in study design, data collection and analysis, decision to publish, or preparation of the manuscript.

### Grant Disclosures

The following grant information was disclosed by the authors:
Hunan Key Laboratory for Internet of Things in Electricity.

### Competing Interests

The authors declare that they have no competing interests.

### Author Contributions

- Jianwei Tian conceived and designed the experiments, performed the experiments, analyzed the data, prepared figures and/or tables, authored or reviewed drafts of the article, and approved the final draft.
- Hongyu Zhu conceived and designed the experiments, performed the experiments, analyzed the data, prepared figures and/or tables, and approved the final draft.

### Data Availability

The data is available in the Supplemental File.

### Supplemental Information

Supplemental information for this article can be found online at http://dx.doi.org/10.7717/peerj-cs.3352#supplemental-information.

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
