# Peer review of "Evaluating the efficacy of AI-driven intrusion detection systems in IoT: a review of performance metrics and cybersecurity threats"

_PeerJ Computer Science, doi:10.7717/peerj-cs.3352_

## Round 0.1 · original submission · Major Revisions

· Academic Editor

Major Revisions

Reviewer 1 ·

Basic reporting

1. The paper needs extensive professional proofreading. Sentences are often verbose or imprecise.

2. The related work is lengthy and citation-dense, yet it reads as a literature dump rather than a focused synthesis. It is better to critically do group studies by topics (e.g., type of AI model, dataset usage, real-world) to better support the stated research gaps.

Experimental design

3. Many citations are used without sufficient explanation of what they contributed. It is better to focus on a smaller number of studies with deeper critical analysis.

4. It better to explain what the PRISMA 2020 guidelines and it is is chosen

5. It’s unclear how comparisons were standardized across studies with different metrics, datasets, and AI models.

6. it is better to reframe this as a qualitative systematic review or include a clearer quantitative synthesis methodology

7. The final set of 51 studies shows geographic bias (heavy dominance by India, Saudi Arabia, and China). It is better to include more studies from different geographic areas with variations of studies.

Validity of the findings

8. It is better to include the following in the summary table (AI techniques, datasets, threat types) not only the systematic literature review paper.

9. It is preferred to include both funnel plot and Egger’s/Begg’s test results to support the claims of publication bias rigorously.

10. In the Meta-Analysis Results by Metric, you mention that AI methods yielded a 21.15% higher accuracy than their traditional counterpart. Please clearly define what “traditional” IDS refers to (e.g., rule-based, signature-based) and specify datasets used in performance comparison.

11. Performance comparison largely relies on accuracy, which is misleading for imbalanced datasets typical in intrusion detection.

Additional comments

12. Claims interest in explainable AI (XAI) is rising but offers no quantitative or qualitative exploration.

13. Terms like "hybrid," "ensemble," or "deep learning" are used loosely without consistent taxonomy or criteria.

·

Basic reporting

The manuscript is written in clear, precise, and professional English, making it accessible to both researchers and practitioners in the field. The authors have done a commendable job of contextualizing the study with a thorough review of relevant literature and contemporary developments in AI-based Intrusion Detection Systems (IDS) for IoT.

The manuscript follows a logical and professional structure, with clearly defined sections. Citations are current and appropriate, covering foundational and recent works from 2021 to 2025. The contour-enhanced funnel plot and meta-regression figures are referenced effectively in the text; however, the inclusion of raw data or a PRISMA diagram showing study selection would strengthen transparency and reproducibility.

The review topic is both timely and relevant, addressing the growing intersection of AI and IoT security. While the field has been reviewed before, this work distinguishes itself by offering a meta-analytic approach that quantitatively evaluates performance trends, filling an important gap in the literature.

The introduction effectively sets the stage for the analysis, clearly articulating the relevance of AI to IDS in IoT settings and outlining the study’s motivation. The audience is well defined, and the scope aligns with the journal’s aims.

Experimental design

The article presents a comprehensive and methodologically sound meta-analysis. The use of effect size estimation, heterogeneity measures (I², Q, τ²), and publication bias diagnostics adds rigor. The incorporation of a meta-regression to assess temporal performance trends is a valuable and rarely seen addition in this domain.

That said, the manuscript would benefit from greater clarity in describing the study selection and inclusion/exclusion criteria. A dedicated section on methodology, possibly including a flow diagram, would improve the transparency and replicability of the review.

The survey methodology is generally robust and covers a wide range of sources. However, expanding the scope to include grey literature (e.g., technical reports or dissertations) may help mitigate the risk of publication bias, which the authors themselves identify through funnel plot asymmetry.

The review is logically structured and thoughtfully organized. Subsections such as AI technique categorization, performance metrics, and practical implications are handled with clarity and coherence.

Validity of the findings

The manuscript provides well-reasoned conclusions that are firmly grounded in the data. The positive performance advantage of AI-based IDS models over traditional approaches is statistically supported and convincingly argued.

Importantly, the authors acknowledge the high heterogeneity across studies and interpret findings with appropriate caution. Their discussion of real-world applicability—particularly the limitations of offline evaluations and the need for interpretability—shows a mature understanding of the complexities involved.

The conclusion also does well to point out current gaps and future research needs, including the push toward explainable AI (XAI), deployment in real-time environments, and adaptive learning to handle evolving threats. These forward-looking insights add depth and practical relevance to the findings.

Additional comments

Overall, the article makes a meaningful and timely contribution to the cybersecurity literature, particularly in the context of IoT networks. Its meta-analytic lens provides valuable quantitative evidence to complement existing narrative reviews.

Expanding the methodological transparency and including a more explicit breakdown of study selection would further strengthen the manuscript. A slightly expanded discussion of deployment-related challenges (e.g., latency, energy constraints, hardware compatibility) would enhance its applicability in real-world settings.

The article strikes a commendable balance between technical depth and accessibility, making it suitable for both academic and industry audiences.

---

## Round 0.2 · accepted · Accept

· Academic Editor

Accept

Dear authors, we are pleased to verify that you meet the reviewer's valuable feedback to improve your research.

Thank you for considering PeerJ Computer Science and submitting your work.

Kind regards
PCoelho

Reviewer 1 ·

Basic reporting

-

Experimental design

-

Validity of the findings

-

Additional comments

Please read the following paper; it might benefit your research:
Almedires, Motaz Abdulaziz, Ahmed Elkhalil, and Mohammed Amin. "Adversarial attack detection in industrial control systems using LSTM-based intrusion detection and black-box defense strategies." Journal of Cyber Security and Risk Auditing 2025, no. 3 (2025): 4-22.